# Emerging Therapeutic Targets and Drug Resistance Mechanisms in Immunotherapy of Hematological Malignancies

**DOI:** 10.3390/cancers15245765

**Published:** 2023-12-08

**Authors:** Wioletta Olejarz, Grzegorz Basak

**Affiliations:** 1Department of Biochemistry and Pharmacogenomics, Faculty of Pharmacy, Medical University of Warsaw, 02-091 Warsaw, Poland; 2Centre for Preclinical Research, Medical University of Warsaw, 02-091 Warsaw, Poland; 3Department of Hematology, Transplantation and Internal Medicine, Medical University of Warsaw, 02-091 Warsaw, Poland; grzegorz.basak@wum.edu.pl

**Keywords:** immunotherapy, CR-T, hematological malignancies, checkpoint inhibitors, TME

## Abstract

**Simple Summary:**

Chimeric antigen receptor T (CAR-T) therapy has revolutionized cancer immunotherapy by inducing a durable response in patients with acute lymphoblastic leukemia (ALL) and non-Hodgkin lymphoma (NHL). The challenges for cancer immunotherapy concern complex resistance mechanisms; therefore, a very important therapeutic approach is to focus on the development of rational combinations of targeted therapies with non-overlapping toxicities. Recent progress in the development of potential therapeutics has significantly improved anticancer responses, while next-generation CAR-T-cells may overcome current limitations and decrease unwanted side effects in targeting hematological malignancies.

**Abstract:**

CAR-T cell therapy has revolutionized the treatment of hematological malignancies with high remission rates in the case of ALL and NHL. This therapy has some limitations such as long manufacturing periods, persistent restricted cell sources and high costs. Moreover, combination regimens increase the risk of immune-related adverse events, so the identification new therapeutic targets is important to minimize the risk of toxicities and to guide more effective approaches. Cancer cells employ several mechanisms to evade immunosurveillance, which causes resistance to immunotherapy; therefore, a very important therapeutic approach is to focus on the development of rational combinations of targeted therapies with non-overlapping toxicities. Recent progress in the development of new inhibitory clusters of differentiation (CDs), signaling pathway molecules, checkpoint inhibitors, and immunosuppressive cell subsets and factors in the tumor microenvironment (TME) has significantly improved anticancer responses. Novel strategies regarding combination immunotherapies with CAR-T cells are the most promising approach to cure cancer.

## 1. Introduction

CAR-T therapy has revolutionized the treatment of hematological malignancies with high remission rates in the case of ALL and NHL [1]. In some contemporary clinical trials, the cure rate of childhood ALL has exceeded 90% [2]. Current therapeutic approaches include inotuzumab, blinatumomab and CAR-T cell therapy for B-ALL, which offer hope for high-risk patients or poor early-treatment responders who do not have targetable genetic lesions [3]. Dual-targeted (CD19/CD22) CAR-T cells show a prominent antileukemia activity in patients with relapsed/refractory ALL (R/R ALL) [4]. Importantly, long-term outcome in children and adults might be increased via treatment of relapsed and/or refractory (R/R) B-ALL with allogeneic stem cell transplant (Allo-SCT) after CAR-T therapy [5]. NHL is the most prevalent group of hematological malignancies, with aggressive or indolent entities. Among patients with R/R NHL, the 5-year survival duration after diagnosis is poor. Thus, treatment of R/R NHL is mainly based on targeted/directed therapies, including B-cell receptor (BCR) signaling inhibitors, checkpoint inhibitors, Bcl-2 inhibitors, monoclonal antibodies (mAbs), immunomodulatory agents, epigenetic modulators and CAR-T cells [6,7]. CAR-T therapy has certain limitations like long manufacturing periods, persistent restricted cell sources and high costs [8]. Moreover, combination regimens increase the risk of immune-related adverse events, so finding new therapeutic targets is key to minimize the risk of toxicities and to guide more effective approaches [9]. A comprehensive management of ALL and NHL is also important to determine the minimal/measurable residual disease (MRD) [10,11]. The identification of MRD that persists after chemotherapy is key when assessing the prognosis of patients with ALL [12]. The diagnosis of MRD after hematopoietic stem cell transplantation (HSCT) is of particular importance in ALL and is important to guide post-transplant maintenance treatment [13]. MRD and immunosurveillance are regulated by many factors in the TME of hematological malignancies [14]. In this review, we describe emerging therapeutic targets and elucidate drug resistance mechanisms in immunotherapy of hematological malignancies.

## 2. Targets for Drugs in Cancer Immunotherapy

### 2.1. Immune Checkpoint

A crucial mechanism by which tumors can escape the anti-tumor immune response is enhanced signaling of the programmed cell death protein 1 (PD-1)/cytotoxic T-lymphocyte antigen-4 (CTLA-4) pathway. Clinical responses in R/R disease in hematological malignancies have been observed after the use of anti-PD-1 therapy [15]. PD-1-blocking antibodies have been used in patients with heavily treated R/R Hodgkin lymphoma to enhance immunity in several malignancies and have obtained durable responses [16]. Immune checkpoints such as CTLA-4 and PD-1 have an important role in the maintenance of peripheral immune tolerance. Then, effective anti-tumor immune responses are caused by the blockade of CTLA-4 and PD-1 [17,18]. Immune checkpoints are also important targets in cancer immunotherapy, such as lymphocyte-activation gene-3 (LAG-3), T cell immunoglobulin and ITIM domain (TIGIT), V-domain Ig suppressor of T cell activation (VISTA), and T cell immunoglobulin and mucin-domain containing-3 (TIM-3) [17] (Figure 1).

Immune checkpoint inhibitors (ICIs) reactivate suppressed T cells, especially in the TME [19]. New agents are being investigated to target immune checkpoints and cancer-intrinsic oncogenic pathways [20]. Immune checkpoint inhibitors (ICIs) have shown promising clinical effects in the treatment of hematological malignancies [21,22]. Ipilimumab, as an anti-CTLA-4 monoclonal antibody, is an FDA-approved immune checkpoint inhibitor [23]. Anti-PD-1 (pembrolizumab, nivolumab and cemiplimab) and anti-PD-L1 (avelumab, atezolizumab and durvalumab) antibodies have been approved for use in the treatment of hematological malignancies and obtained long-term efficacy [24,25]. ICIs targeting CTLA-4 and the PD-1/PD-L1 axis strengthen anti-tumor immune responses by disrupting co-inhibitory T-cell signaling [26]. These antibodies acting against immune checkpoints are very effective for cancer immunotherapy of hematological malignancies [27]. ICIs as first-line therapies for advanced cancers have led to unprecedented results in patients with previously incurable metastatic diseases [28]. Sustained overexpression of co-inhibitory receptors on CD8+ T cells promotes T-cell exhaustion or dysfunction, leading to cancers, whereas autoimmunity is caused by dysregulated expression of co-inhibitory receptors on CD4+ T cells [29].

Also, the costimulatory molecule 4-1BB/CD137 or OX40/CD134 agonistic antibodies cause T-cell priming via dendritic cell activation [30]. Glucocorticoid-induced tumor necrosis factor receptor-related protein (GITR) as a costimulatory receptor plays an important role in regulating the effector functions of T cells. Bispecific molecules composed of an anti-PD-1 antibody linked with a multimeric GITR ligand (GITR-L) show dose-dependent tumor growth inhibition [31].

### 2.2. Clusters of Differentiation (CDs) and B-Cell Maturation Antigen (BCMA)

Clusters of differentiation like CD19, CD20, CD30, CD33, CD38, CD47, CD123, CD138 and CD269 (BCMA) as targets have demonstrated great potential for CAR-T cell therapy, while CD23 and SLAMF7 have also shown promising results in clinical trials [32,33] (Figure 1).

CD20

Rituximab is an anti-CD20 chimeric monoclonal antibody used in patients with various CD20-expressing lymphoid malignancies [34]. It was shown that rituximab has improved results in patients with B-cell non-Hodgkin lymphomas [35]. Rituximab not only prolongs the time to disease progression but also extends overall survival, as demonstrated in clinical trials [34]. Rituximab is well tolerated; however, increased use of rituximab has been associated with hypersensitivity reactions (HSRs) [36].

CD25

CD25 is widely expressed on regulatory T cells (Tregs) and activated circulating immune cells, as well as in hematological malignancies. Infiltration of Tregs in the tumor microenvironment (TME) is associated with the progression of cancers, but Teff/Treg cell ratio shows an efficient anti-tumor response to immunotherapy [37]. Camidanlumab tesirine (ADCT-301) targeting human CD25, either alone or in combination with ICIs, is being investigated in a phase I trial (NCT03621982) [38].

CD30

CD30 is a transmembrane protein from the tumor necrosis factor receptor superfamily and is expressed on activated T and B lymphocytes and in various lymphoid neoplasms [39]. The CD30 antigen is highly expressed on neoplastic cells in hematological malignancies such as DLBCL, representing an ideal immunotherapeutic target [40]. CAR-T cells targeting CD30 in patients with R/R CD30+ hematological malignancies have shown high response rates with durable remissions [41]. A CD30-directed antibody–drug conjugate (ADC), brentuximab vedotin, is approved for treating patients with CD30-expressing hematological malignancies [42].

CD33

Lintuzumab-CD28/CD3ζ CD33 CAR-T immunotherapy is now under evaluation in a preclinical trial involving children and adolescents/young adults with relapsed/refractory acute myeloid leukemia (AML) [43]. In an ex vivo study against AML cells, targeting CD33 showed AMG 330-mediated T-cell cytotoxicity and expansion [44]. Gemtuzumab ozogamicin has been approved as a selective anti-CD33 antibody–calicheamicin conjugate for the treatment of hematological malignancies [45].

CD38

The transmembrane glycoprotein CD38 has a comparatively low expression on normal and lymphoid cells and is expressed in high levels in multiple myeloma (MM) [46,47]. Daratumumab as a monoclonal antibody (mAb) targeting CD38, CD38-specific CAR-T cells and bispecific antibodies to stimulate T cells to eliminate CD38+ MM cells is a very effective therapeutic approach [48]. Its synergistic activity with pomalidomide and lenalidomide, as well as PD1/PD-L1 inhibitors and CD38-targeting antibodies, has been demonstrated in preclinical and clinical studies [49].

CD47

CD47 is expressed on healthy and malignant cells and may regulate macrophage-mediated phagocytosis by sending a “don’t eat me” signal to the signal regulatory protein alpha (SIRPα) receptor [50]. High levels of CD47 in hematological malignancies are associated with mechanisms of immune evasion [51]. Many studies have confirmed that blocking CD47 interaction with SIRPα can enhance cancer cell clearance via macrophage inhibition of CD47/SIRPα interaction, which may increase antigen cross-presentation, leading to an adaptive anti-tumor immune response with T-cell priming [52]. It has been shown that CD47 expression is crucial for the effectiveness of CAR-T therapy [53]. Blocking CD47/SIRPα signaling enhances the anti-tumor effect of CAR-T cells [54].

CD123

CD123 is the interleukin-3 receptor alpha chain (IL-3R). It is expressed on more differentiated leukemic blasts and leukemic stem cells (LSCs), and is widely overexpressed in B-ALL and other hematological malignancies. CD123 is an attractive therapeutic target for treating AML or blastic plasmacytoid dendritic neoplasm (BPDCN). Different anticancer agents, like Tagraxofusp (SL401, Stemline Therapeutics), that are directed against CD123 have demonstrated promising results for the treatment of either R/R disease or MRD [55].

CD138

Synergistic cytotoxicity in hematological malignancies is triggered by combinations of the proteasome inhibitor carfilzomib (CFZ) with a pharmacological isocitrate dehydrogenase 2 (IDH2) inhibitor (AGI-6780). CFZ/AGI-6780 treatment enhances the death of primary CD138+ cells in multiple myeloma (MM) patients and presents a beneficial cytotoxicity profile toward bone marrow-derived stromal cells and peripheral blood mononuclear cells (PBMCs) [56].

BCMA

B-cell maturation antigen (BCMA), as a transmembrane glycoprotein in the tumor necrosis factor receptor superfamily 17 (TNFRSF17), is an antigen expressed on the surface of plasma cells [57]. BCMA is expressed on malignant plasma cells (PCs) and is a key target for multiple myeloma (MM) treatment [58]. Antibody–drug conjugates (ADCs), like Belantamab mafodotin-blmf (GSK2857916), as BCMA-targeted therapeutics have been approved for highly refractory MM. BCMA-targeted ADCs have achieved remarkable clinical responses in patients with R/R MM, including bispecific T cell engagers (BiTE) conjugated to CAR-T cells [59].

### 2.3. Signaling Pathways

#### 2.3.1. Phosphatidylinositol 3-kinase (PI3K)/AKT/Mammalian Target of the Rapamycin (mTOR) Signaling Pathway

PI3K/Akt/mTOR signaling plays important roles in promoting tumor initiation, progression and therapy responses [60]. This pathway may be activated in childhood ALL, as well as in some lymphoproliferative disorders and pediatric lymphomas [61]. mTOR is a protein kinase regulating metabolism, cell growth, survival and immunity. mTOR catalyzes the phosphorylation of AKT, protein kinase C (PKC), ribosomal protein S6 kinase β-1 (S6K1), eukaryotic translation initiation factor 4E-binding protein 1 (4E-BP1) and type-I insulin-like growth factor receptor (IGF-IR). Activation of mTOR may promote tumor growth and metastasis [45]. IFN may induce functional HIF-1α expression and regulate epithelial–mesenchymal transition (EMT), cellular metastasis and anti-apoptosis activity via the activation of the PI3K/AKT/mTOR axis. Moreover, targeting HIF-1α is important for inhibiting tumorigenesis and EMT [62]. It has been shown that PI3K/AKT/mTOR inhibitors may improve patient response in cancers [60]. Rhapontigenin (Rha) may inhibit the progression of cancer by disrupting angiogenesis and EMT [63]. Also, sirolimus as a target in this pathway has been used successfully in pediatric hematological malignancies [61].

It has been shown that poor prognosis in hematological malignancies is associated with the PI3K pathway, whereas a critical mechanism underlying PI3K inhibitor resistance in lymphoma is IL-6-induced STAT3 or STAT5 activation [64,65].

#### 2.3.2. The Transforming Growth Factor (TGF)-β Signaling Pathway

TGF-β is a cytokine that signals via plasma membrane TGF-β type I and type II receptors and intercellular SMAD transcriptional effectors [66]. TGF-β, in the early phase of tumorigenesis, has tumor suppressive functions through apoptosis and cell cycle arrest [67], whereas in late-stage cancer, TGF-β can promote tumorigenesis, metastasis and chemoresistance [68]. Through the production of mitogenic growth factors, tumor cells develop mechanisms to overcome the TGF-β-induced suppressor effects and stimulate tumor proliferation and survival [69]. TGF-beta has a dual role as both a tumor suppressor and a pro-oncogenic factor; therefore, the choice of therapeutic drug dosage and patient selection should be careful. Moreover, members of the TGF-β signaling pathway are being considered as key molecular targets for prevention and treatment of cancer and metastasis [68].

#### 2.3.3. Signal Transducer and Activator of Transcription 3 (STAT3)

STAT3 controls autophagy molecules and induces apoptosis in T lymphocytes [70]. STAT3 is a crucial molecular hub in malignant tumors that plays important roles in promoting the production of immunosuppressive factors and inhibiting the expression of critical immune regulators [71]. STAT3 is key target in cancer immunotherapy, which is important in enhancing anti-cancer immune responses by rescuing the suppressed immunologic microenvironment in tumors [72]. Targeting STAT3 in combination with other drugs may prove to be a successful therapeutic strategy to overcome acquired DR [73].

#### 2.3.4. Mitogen-Activated Protein Kinase (MAPK) Signaling Pathways

MAPK pathways involving extracellular signaling-regulated kinases (ERKs), the Jun N-terminal kinases (JNK) and p38 MAPK are signal transduction pathways important for modulating drug sensitivity and resistance in cancers [74]. It has been confirmed that the RAS/RAF/MEK/ERK (MAPK) signaling cascade is key for cell inter- and intra-cellular communication, which regulates cell functions such as differentiation, growth and survival [75]. Acquired resistance and insensitivity to drug treatment may be caused by a number of genetic and epigenetic alterations in MAPK signaling; therefore, they are key targets for cancer immunotherapy [76]. The Ras/Raf/MEK/ERK pathway is associated with sensitivity and resistance to leukemia therapy [77]. Growth factors and mitogens may use the Ras/Raf/MEK/ERK signaling cascade to regulate gene expression and prevent apoptosis. Mutations occur in the components (e.g., Ras and B-Raf) of these pathways and in genes encoding upstream receptors (e.g., EGFR and Flt-3); moreover, chimeric chromosomal translocations (e.g., BCR-ABL) transfer their signals via these pathway [78]. Also, granulocyte/macrophage colony-stimulating factor (GM-CSF) and cytokine genes have trans-acting binding sites for the transcription factors regulated by this cascade [79,80].

### 2.4. Exhaustion and Senescence

Exhausted T cells contain heterogeneous cell populations with unique differentiation and functional states and have altered effector functions (cytokine production and killing function), dysregulated metabolism-altered signaling cascades and poor memory recall response [81,82]. These cells lose their effector functions, exhibit altered epigenetic signatures and transcriptional networks, and gain the constitutive expression of coinhibitory receptors [83]. It has been shown that antibodies targeting PD-1/PD-L1 reinvigorate “exhausted” T cells in TME and show persistent response, durable remission and acceptable toxicity profile [84]. The heterogeneity comprises stem-like and terminally differentiated cells within the exhausted CD8+ T cell lineage [85]. Environmental signals promote epigenetic alterations, which set the transcriptomes needed for T cell function [86].

Immunosenescence is a complex regulatory process with changes in both the innate and adaptive immune responses and might be caused by an increased activity of immunosuppressive cells [87,88]. It is associated with thymic involution, naïve/memory cell ratio imbalance, epigenetic changes, chronic inflammation and dysregulated metabolism and is considered the main risk factor for age-related diseases [89]. Chronic antigen stimulation causes premature senescence of immune cells, with a proinflammatory senescence-associated phenotype [89]. This process is induced by damage signals such as oxidative stress, mitochondrial dysfunction, persistent DNA damage and cytokines [90].

Importantly, EVs derived from effector CAR-T cells have anti-tumor potential in treating hematological malignancies [91]. These vesicles may have secretory phenotypes associated with exhaustion and senescence [92,93]. EVs like exosomes derived from exhausted CD8+ T cells have distinct lncRNA expression profiles and could attenuate the function of CD8+ T cells [94].

### 2.5. The Immunosuppressive Tumor Microenvironment (TME)

By increasing T regulatory (Treg) cells and inhibiting T effector (Teff) cell function, tumor-extrinsic factors may alter the composition and activity of tumor-infiltrating lymphocytes (TILs) and promote tumor progression. The response rates and clinical outcomes of anti-cancer therapy are limited by factors like soluble suppressive molecules, immunosuppressive cells or inhibitory receptors expressed by immune cells [95]. In the TME, tumor cells escape anti-tumor responses by constantly evolving to reduce neoantigen generation [95]. Tumor-associated macrophages (TAMs), myeloid-derived suppressor cells (MDSCs), Treg cells, Breg cells and tumor-associated neutrophils (TANs) play a crucial role in shaping the tumor immunosuppressive environment. They can inhibit the function of effector cells, such as CD8+ T cells, NK cells and Th1 cells, by releasing cytokines, such as TGF-β, IL-10, IL-35 and adenosine, and expressing PD-L1 [87,96,97]. Treg cells are heterogeneous and express different immune checkpoint molecules, which play critical roles in the maintenance of immune homeostasis by favoring their immune suppressive function [98,99]. Myeloid-derived suppressor cells (MDSCs) play a critical role in the regulation of the immune response in cancer [100,101]. Direct targeting of MDSCs may abrogate their pro-tumorigenic impact within the tumor microenvironment through the activation of the adaptive immune response [100]. The immunosuppressive network also includes polymorphonuclear (PMN-MDSC) myeloid-derived suppressor cells, which are immature myeloid cells induced by inflammatory mediators and monocytic MDSC (M-MDSC) [87]. Quiescent cancer cells (QCCs) form clusters with higher tumorigenic capacity and constitute immunotherapy-resistant reservoirs [102].

## 3. New Emerging Targets in Cancer Immunotherapy

Antigens such as CD27, CD37, CD70, CD80, CD86, B7-H3 and B7-H4, which are expressed in hematological malignancies and targeted by CAR-T cells, are promising candidates for clinical development [103,104,105] (Figure 2).

CD27 and CD70

CD27 is a member of the tumor necrosis factor receptor superfamily. CD27, by binding to its natural ligand CD70, provides a costimulatory signal important in T-cell activation [106]. Varlilumab, as a CD27 agonistic antibody, has shown promising efficacy in hematological malignancies, particularly in combination approaches with PD1 axis-targeting ICIs, such as atezolizumab and nivolumab [107]. Importantly, CD70 may be used as a target for antibodies inducing ADCC and specific dendritic cell vaccination [106,108].

CD37

CD37 is a cell-surface tetraspanin expressed on B-cells but is absent on normal stem cells and plasma cells. Anti-CD37 monoclonal antibodies improve the overall survival of patients with NHL and chronic lymphocytic leukemia (CLL) [109,110]. Also, chemical agents like anti-CD37 antibodies are a promising therapeutic approach in B-cell malignancies. Furthermore, novel antibodies targeting CD37 are being evaluated in clinical trials [111,112,113].

CD80 and CD86

T cells are regulated by B7 ligands (B7.1 or CD80 and B7.2 or CD86), which are expressed on antigen-presenting cells (APCs) like dendritic cells (DCs) and interact with CD28 and CTLA-4 receptors on T cells [114]. T-cell function is associated with the PD-L1/PD-1 axis. Also, DCs are an important target of PD-L1-blocking antibodies. It has been shown that blocking PD-L1 on DCs enhances T-cell priming via B7.1/CD28 interaction [115]. Therefore, B7 ligands are key targets in cancer immunotherapy [116].

B7-H3 and B7-H4

B7-CD28 family member proteins like B7-H3 and B7-H4 are inhibitory B7 family checkpoint molecules [117]. B7-H3 (CD276) is induced on APCs and plays an important role in the inhibition of T-cell function. B7-H3 is a key target for cancer immunotherapy through its expression on tumor cells and correlates with poor clinical outcomes [118,119]. Also, the B7-H4 molecule may promote immune escape by inhibiting the cycle of T cells, proliferation and cytokine secretion [120]. Clinical studies have confirmed that B7-H4 is a target for cancer immunotherapy [121,122].

## 4. Extracellular Vesicles (EVs)

EVs like exosomes are key in cancer immunotherapy because of their unique composition profiles. Their components, like DNA, mRNA, microRNA, long noncoding RNA, circular RNA and proteins, play a crucial role in regulating tumor growth and metastasis [123]. EVs influence the phenotype and immune-regulation functions of targeted cells by delivering their cargos to the targeted cells. Tumor-derived exosomes can induce apoptosis of activated CD8+ T cells, suppress NK cell activity and inhibit immune cell proliferation [124]. A growing body of evidence confirms that exosomes, by transporting numerous pro-angiogenic biomolecules like vascular endothelial growth factors (VEGFs), matrix metalloproteinases (MMPs) and microRNAs, may participate in cancer angiogenesis and progression [125]. EVs carry bioactive molecules from the originating cells and interact with immune cells, stromal cells and endothelial cells in the TME [126]. They can be considered as predictive biomarkers, drug carriers and therapeutic targets in hematological malignancies [127,128,129].

## 5. Mechanisms of Cancer Immune Resistance

Tumor heterogeneity; tumor antigen and major histocompatibility complex (MHC) modulation; immunosuppressive cell subsets and factors in the tumor microenvironment (TME); anti-apoptotic pathways and T cell activation-induced cell death (AICD); and checkpoint inhibitory ligands are associated with cancer immune resistance [130] (Figure 3).

### 5.1. Tumor Heterogeneity

Intratumor heterogeneity is a key challenge in cancer medicine. It has been shown that tumor contains diverse populations of subclones with a wide range of genetic and epigenetic differences [131]. In a process called subclonal cooperativity, tumor may produce signaling factors that increase the tumorigenicity and growth capacity of neighboring tumor cells. These cells with acquired resistance mechanisms and self-renewing properties are capable of escaping from immune surveillance and often survive treatment [132,133].

### 5.2. Tumor Antigen and Major Histocompatibility Complex (MHC) Modulation

MHC class I molecules bind peptides and then transport and display this antigenic information to CD8 T cells [134]. A loss of MHC/HLA class I molecules is the main mechanisms of tumor immune escape from T-cell recognition and destruction [135]. Carcinoma cells lose their expression of MHC class I molecules and tumor antigens and have low immunogenicity. By contrast, a loss of MHC class II molecules may promote immune evasion by affecting the priming of APCs and CD4+ Th cells [136]. In patients with diffuse large B-cell lymphomas (DLBCLs), the MHC class II genes (e.g., HLA-DRA) correlated with better survival. It was confirmed that the loss of tumor immunosurveillance had a negative effect on patient outcome in DLBCL. In MHC class II-negative cases, fewer tumor-infiltrating CD8(+) T cells were detected [137,138]. Both CD4+ T and CD8+ T cells are effective in tumor immunotherapy by causing directed immune attack against tumor cells [139,140].

### 5.3. Immunosuppressive Cell Subsets and Factors in the Tumor Microenvironment (TME)

Cancer immunoediting occurs through three phases termed elimination, equilibrium and escape, rendering a tumor less immunogenic and more capable of establishing an immunosuppressive microenvironment [141]. As mechanisms of resistance to immunotherapy, tumor immunogenicity and the immunosuppressive environment enable disease progression [142]. The resistance mechanisms are complex and highly heterogeneous, and the TME is a major location for resistance to occur [143]. Overcoming tumor-induced immune suppression can increase the efficiency of immunotherapies through immune-mediated tumor clearance [144].

Tumor cell interactions with the TME are important in multidrug resistance (MDR). By recruiting immune cells, TME-induced factors secreted by cancer cells and cancer-associated fibroblasts (CAFs) create an inflammatory microenvironment. MDSCs and inflammatory tumor-associated macrophages (TAMs) enhance chronic inflammation, which nurtures tumor-initiating/cancer stem-like cells (CSCs) and induces both EMT and MDR, leading to tumor relapses [145]. MDSCs, TAMs and CAFs via inflammatory cytokine and chemokine secretion are involved in EMT and MDR. Furthermore, the cytokine content in the TME may be caused by tumor cell differentiation and T-cell resistance [146]. Immune modulatory compounds, such as demethylating agents, mTOR inhibitors and low-dosed histone deacetylase inhibitors, may decrease MDR by targeting the inflammatory process [145]. Also, by eliciting immune system cells and initiating acute inflammation that leads to cancer destruction, immune-based therapies may modulate the tumor microenvironment [147]. The development of treatment resistance is associated with a hypoxic environment and metabolic derangements in the TME [148].

Hypoxia-inducible factor-1 alpha (HIF-1α) is a transcription factor that, in an oxygen-free environment, upregulates genes involved in cancer progression [149,150]. Growing evidence indicates that hypoxic cancer cells can adapt metabolically to HIF-1 inhibition, which causes drug resistance [151]. It has been confirmed that drug resistance is caused by an expansion of both stroma cells and cancer cells, which causes hypoxia due to HIF-1α activation and, therefore, provides a novel target for tumor therapy [62,152].

Immunomodulatory drugs (IMiDs) like thalidomide, pomalidomide and lenalidomide have anti-tumor activity in the TME [153]. Avadomide, a next-generation cereblon E3 ligase modulator (CELMoD), shows anti-tumor potential in the TME and promising clinical efficacy in DLBCL. Moreover, lenalidomide in a combination therapy with rituximab is promising for the treatment of relapsed/refractory (R/R) FL [154].

### 5.4. Anti-Apoptotic Pathways and T Cell Activation-Induced Cell Death (AICD)

Apoptosis is a form of programmed cell death that leads to the efficient removal of damaged cells. Alterations in apoptosis mechanism involve Bcl-2 family proteins, cell cycle and repair system dysregulation, inhibitors of apoptosis as inhibitory proteins (IAPs), tumor suppressor (p53) regulation and cell progression pathway activation of NF-κB [155]. Apoptosis dysregulation is responsible for tumor development, while defects in the death pathways may limit the efficacy of therapies and result in drug resistance [156]. In hematological malignancies, aberrations of the intrinsic and extrinsic apoptotic pathways have been identified to be associated with cancer progression and drug resistance [157]. Dysregulation of the B-cell leukemia/lymphoma-2 (BCL-2) family of proteins in the intrinsic apoptotic pathway is observed in hematologic malignancies; therefore, targeting the apoptotic pathways like the regulatory BCL-2 family in the intrinsic pathway is an important therapeutic approach in these patients [158,159,160].

AICD of T cells is a process for regulating the immune system, where overactivated harmful T cells are eliminated via Fas/FasL-mediated apoptosis [161]. AICD is caused by the stimulation of activated T cells, which results in their apoptosis through the Fas/Fas ligand (FasL) interaction that is predominantly involved in AICD of CAR-T cells [162,163,164]. It has been shown that the AICD mechanism is key to improve the anti-tumor effect of CAR-T cells, whereas AICD resistance is associated with a reduced surface expression of CD95L upon restimulation [162,165,166,167].

### 5.5. Checkpoint Inhibitory Ligands

Immune checkpoint blockade therapies have improved treatment for many types of tumors. Despite the promising long-term responses, the majority of patients fail to respond to drugs and show primary resistance [168]. CTLA-4 and PD-1 receptors have a key role in regulating T cell responses [169]. Also, LAG-3, TIGIT, Tim-3 and VSTA are immune checkpoint inhibitors that play an important role in regulating T-cell responses and maintaining immune homeostasis [170]. LAG-3 and PD-1 molecules synergistically regulate T-cell function to promote tumoral immune escape [171]. Immune escape from PD-L1/PD-1-targeted therapy includes the downregulation of the major histocompatibility complex in cancer cells, the minimal activation of cancer-specific T cells, the poor infiltration of T cells into tumors, the lack of strong cancer antigens or epitopes recognized by T cells, and the presence of immunosuppressive factors and cells in the tumor microenvironment [172]. The “primary” activation of multiple oncogenic signaling and the “secondary” induction by inflammatory factors such as IFN-γ may be caused by the high expression of PD-L1 in the tumor microenvironment (TME) [84]. In cancers, strong chronic antigenic stimulation via TCR leads T cells to a state of exhaustion through inhibitory immune checkpoint molecules, such as PD-1 and CTLA-4 [173]. CTLA-4 serves as a dominant off-switch, but its combination with anti-PD-1 blockade and multiple checkpoint receptors, including TIM-3 and TIGIT, showed superior patient outcomes [170]. TIGIT expression is upregulated in Tregs and CD8+ cells. It was shown that using a dual therapy of anti-TIGIT and anti-PD-1 significantly improved survival compared to the control and monotherapy groups [174]. By suppressing Teff cell functions, Tregs regulate immune homeostasis and may contribute to cancer development and progression. Tregs mediate the mechanisms of acquired resistance to immune checkpoints (ICIs) within the TME by upregulating apoptotic Treg-induced immunosuppression and immunosuppressive molecules [140].

A high tumor mutational burden (TMB) plays an important role in anti-tumor immunity, with an accompanying elevated neoantigen expression. It has been shown that poorly immunogenic tumors with a low TMB are more resistant to treatment with checkpoint inhibition [175].

Targets in immunotherapy and therapeutic strategies to overcome drug resistance are presented in Table 1.

## 6. EVs in the Tumor Microenvironment as Mediators of Cancer Therapy Resistance

EVs derived from immune cells and tumor cells exhibit unique composition profiles. They can participate in cellular processes by suppressing or promoting cancer [180]. EVs derived from both stromal and tumor cells play an important role in all stages of cancer development [127]. They are mediators of cell–cell communication and may be related to TME-dependent therapy resistance [181]. EVs can potentiate or oppose the development of an aggressive TME, influencing tumor progression and clinical outcome. They are key mediators of immunoregulation in cancer. EVs may support or restrain immunosuppression of lymphoid and myeloid cell populations in tumors by delivering a large number of signals to recipient cells [182]. Cancer cells release exosomes into the circulation to counter anti-tumor immunity systemically via T cells, but EVs secreted by nontumor cells in the TME can also exert immunosuppressive effects [183]. Importantly, cancer-derived exosomes may transfer functional PD-L1 and inhibit immune responses [184]. The potential of exosomes is huge in the field of cancer immunotherapy. They may serve as predictive biomarkers, drug carriers and therapeutic targets to stimulate an anti-cancer immune response [129,185].

Immunostimulatory and immunosuppressive effects of EVs is presented in Figure 4.

## 7. Conclusions and Future Directions

Personalized immunotherapy is the most promising approach to cure cancer and to induce a durable response in patients. CAR-T therapy has revolutionized cancer immunotherapy, especially in hematological cancers, by improving overall survival. Despite progress in clinical investigations, there is still a need to focus on important questions and resolve difficulties associated with current basic understanding and clinical responses. The challenges facing cancer immunotherapy concern complex resistance mechanisms in the development of effective treatment strategies, including the lack of confidence in translating pre-clinical findings and recognizing the optimal combinations of immune-based therapies for individual patients. Also, a very important issue is the deciphering of the complex interactions between the immune system and cancer and the development of improved treatment options [186]. Cancer cells employ several mechanisms to evade immunosurveillance, which causes resistance to immunotherapy; therefore, it is very important for therapeutic approach to focus on the rational combinations of targeted therapies with non-overlapping toxicities. Recent progress in the development of new therapeutics has improved anticancer responses. Studies exploring novel CAR-T products may improve treatment efficacy and reduce toxicities [187]. Next-generation CAR-T-cells may overcome current limitations and decrease unwanted side effects in targeting hematological malignancies. Moreover, signaling pathway blocking could provide a new strategy for successful CAR-T immunotherapy. Also critical in the development of new immunotherapeutic strategies against Treg-mediated acquired resistance mechanisms are CDs as potential targets for the depletion of Tregs. They not only serve as predictive indicators of therapeutic responses to inflammation-associated diseases but also as biomarkers for diagnosis and follow-up. Combination therapies targeting a single kinase component, including PI3K/Akt/mTOR and MAPK inhibitors, have improved patient responses and clinical outcomes. Inhibition of the PI3K/mTOR pathway is a promising therapeutic approach in patients with ALL [188]. Furthermore, targeting of TGF-β signaling modulators and downstream effectors can be a very effective strategy. With an expanding therapeutic armamentarium, it is very important to focus on the rational combinations of targeted therapies. The key in the selection of therapeutic targets is the identification of tumor-promoting factors in the TME. Moreover, targeting EMT and inflammatory pathways by means of immune modulatory compounds, such as demethylating agents, mTOR inhibitors and low-dosed histone deacetylase inhibitors, may reverse MDR. Additional evidence is still needed to better characterize the benefits of the use of CAR-T cells in patients with hematological malignancies. In the future, CAR-T cell therapy may have the potential to be a mainstream therapeutic choice for the elimination of tumors with superior safety and efficacy. Additionally, elucidating how tumors escape from immune checkpoint receptor-targeted therapy is an important issue in the development of new immune agents. Importantly, EVs can be utilized in immune therapy for controlling cancer progression. Additionally, EVs have implications for diagnosis and development of novel therapeutic strategies in hematological malignancies. Identifying predictive biomarkers and designing rational combination therapy are priorities to reduce drug resistance and improve long-term clinical efficacy in cancer immunotherapy.

## Figures and Tables

**Figure 1 cancers-15-05765-f001:**
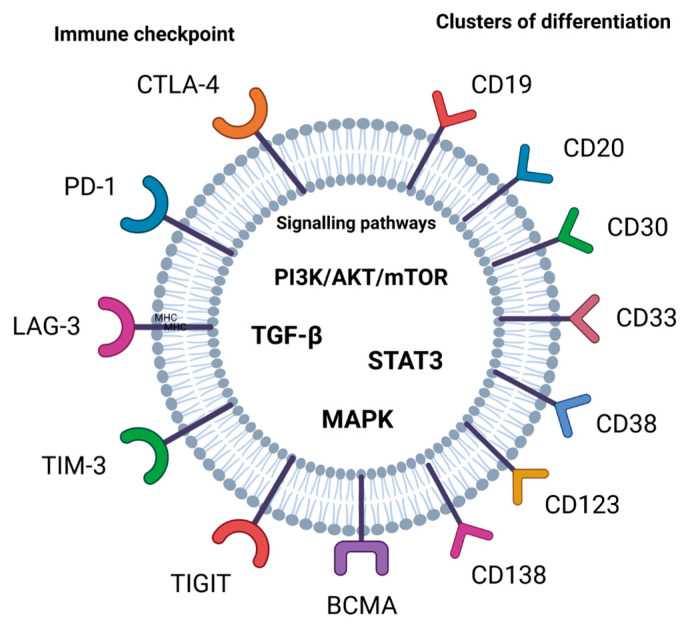
Targets for drugs in cancer immunotherapy.

**Figure 2 cancers-15-05765-f002:**
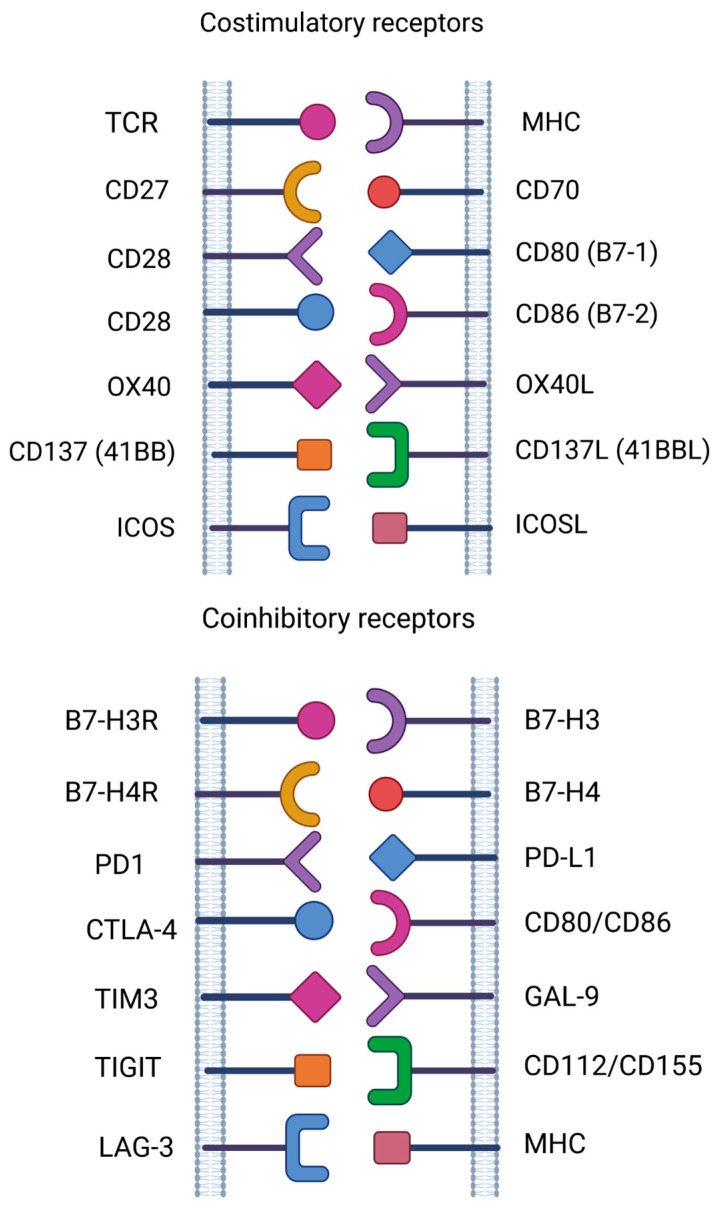
Costimulatory and coinhibitory receptors as targets for cancer immunotherapy of hematological malignancies.

**Figure 3 cancers-15-05765-f003:**
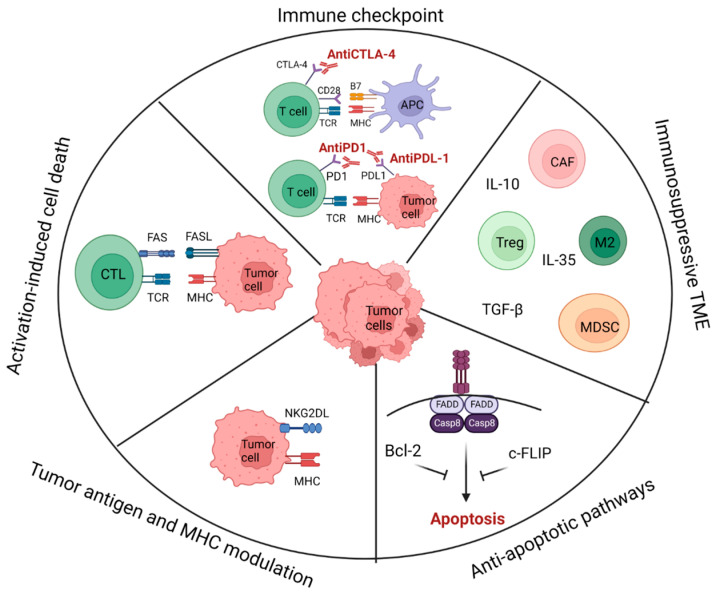
Mechanisms of cancer immune resistance [130].

**Figure 4 cancers-15-05765-f004:**
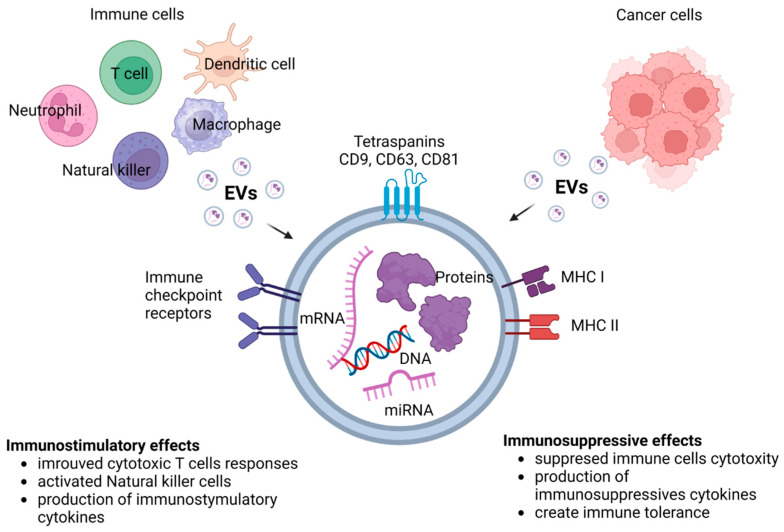
Immunostimulatory and immunosuppressive effects of EVs.

**Table 1 cancers-15-05765-t001:** Therapeutic strategies to overcome drug resistance.

Targets in Immunotherapy	Examples	Drugs	Mechanism of Action	References
Immune checkpoint	CTLA-4 PD-1 LAG-3 TIM-3 TIGIT	Ipilimumab, Nivolumab, Pembrolizumab, Avelumab, Cytarabine, Decitabine, Darubicin, MBG453, Tebotelimab (MGD013)	Reverses T-cell exhaustion. Enhances T-cell activation and effector functions. Broadens TCR repertoire.	[176]
CD and BCMA	CD19 CD20 CD30 CD33 CD38 CD47 CD123 CD138 BCMA	Rituximab, Blinatumomab, Magrolimab (Hu5F9-G4), Epcoritamab, AMG330, Vixtimotamab (AMV564), Vibecotamab (XmAb14045), Flontetuzumab, AMG427, Brentuximab vedotin Belantamab	Activates effector cells. Upregulates proinflammatory cytokines. Increases genomic instability. Modulates metabolic response.	[176,177,178,179]
Signaling pathways	PI3K/Akt/mTOR TGF-β STAT3 MAPK	Rapalog, Rhapontigenin, Trastuzumab, Dabrafenib, Trametinib	Blocks signaling pathways. Decreases inhibitory cytokine production. Enhances T-cell effector function.	[60,61,62,73,74]
Immunosuppressive TME	MDSC Treg CAF TAM	Thalidomide, Lenalidomide, Pomalidomide, Avadomide	Depletes suppressive cells. Redirects cytotoxic effector cells to the TME.	[145,148,153,154]

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
