# Peer review of "Emerging Therapeutic Targets and Drug Resistance Mechanisms in Immunotherapy of Hematological Malignancies"

_cancers, 2023, doi:10.3390/cancers15245765_

Round 1

Reviewer 1 Report

Comments and Suggestions for Authors

The authors of the review entitled " Emerging Therapeutic Targets and Drug Resistance Mechanisms in Immunotherapy of Hematological Malignancies" have summarized a notable number of recent discoveries that have given a turning point in the treatment of leukemias and lymphomas, especially for those cases refractory to therapies.

In particular the authors discussed the need of the identification of therapeutic targets to overcome the risk of immune-related adverse events increases with combination regimens in CAR-T cells immunotherapies.

They highlighted the emerging therapeutic targets and elucidate drugs resistance mechanism in immunotherapy of hematological malignancies, hoping for novel therapeutic strategies by combination immunotherapies with CAR-T cells as approach to cure cancer.

There are some stylistic imperfections and spelling errors through the manuscript, however I think that the topic has been covered comprehensively, hence it deserves publication.

Comments on the Quality of English Language

The quality of English is good, there are some spelling errors.

Author Response

Response: Thank you very much for review our manuscript and positive opinion. We have corrected stylistic imperfections and spelling errors through the manuscript.

Reviewer 2 Report

Comments and Suggestions for Authors

The manuscript titled: Emerging Therapeutic Targets and Drug Resistance Mechanisms in Immunotherapy of Hematological Malignancies

Submitted by Drs Olejarz and Basak, it is a review with current and relevant content. However, it requires some changes to emphasize the relevance of the manuscript:

In figure 3 check the word activation.

In section 2.3 signaling pathways.

The relevance of the aforementioned strategies is not very clear. It would be desirable to include other relevant pathways such as MAPKs. And other factors that influence tumor growth and persistence such as IL-6 and other growth factors.

Author Response

Response: Thank you very much for review our manuscript, positive opinion and important comments.

Our response in manuscript is shown in red text.

In figure 3 check the word activation.

Response: We have corrected the mistake

In section 2.3 signaling pathways.

Response: We have added information about MAPKs, IL-6 and growth factors in section 2.3 signaling pathways.